# Assessing the Impacts of Chloride and Sulfate Ions on Macroinvertebrate Communities in Ohio Streams

## Robert Miltner

Ohio Environmental Protection Agency, Columbus, OH 43216-1049, USA; robert.miltner@epa.ohio.gov

**Abstract:** Salinization of freshwaters is a growing concern, especially in urban catchments. Existing aquatic life criteria for chloride (230 mg/L; a US standard) or total dissolved solids (1500 mg/L; specific to Ohio) do not protect sensitive species, and standards for sulfate have yet to be promulgated on the national level. To help identify water quality thresholds for protection and restoration, species sensitivity distributions were compiled for chloride and sulfate based on field observations of macroinvertebrate communities co-located with water quality samples obtained from rivers and streams throughout Ohio. Additionally, attainment of biological benchmarks for macroinvertebrate communities found in headwater streams were modeled against chloride and sulfate using Bayesian logistic regression. The hazard concentration based on statewide data for chloride was 52 mg/L. The hazard concentration for sulfate based on data from the Western Allegheny Plateau ecoregion was 152 mg/L. The median effect levels from logistic regression for chloride and sulfate varied by ecoregion. Mayfly taxa were disproportionately represented in taxa comprising the lower 5th percentile of the species sensitivity distributions for chloride. However, logistic regression models of individual taxa response (as presence/absence) revealed that some taxa considered sensitive to pollution in general were highly tolerant of chloride. For 166 taxa showing directional response to chloride, 91 decreased and 75 increased. For the 97 individual taxa showing directional responses to sulfate, 81 decreased. Of the 16 taxa showing an increase, 6 are considered tolerant of pollution, 9 facultative and 1 moderately intolerant, the latter being taxa in the dipteran family Tipulidae. The hazard concentrations are useful as protective thresholds for existing high-quality waters. The logistic regression model of attainment can be used to inform management goals conditional on site-specific information.

**Keywords:** chloride; sulfate; salinization; macroinvertebrates; rivers; Bayesian

## 1. Introduction

Dissolved ions in freshwater environments are often collectively expressed as total dissolved solids (TDS mg/L) or measured as specific conductance (SC uS/cm, specific to 25 °C). Typical ranges for relatively undisturbed freshwater streams in Ohio are 160–660 mg/L TDS, and 206–960 μS/cm. Because freshwater organisms are internally saltier than the water they live in, they must actively maintain their internal ionic concentrations and ionic compositions through osmoregulation. Most freshwater organisms are facultative with respect to external osmotic pressure as an adaptation to natural seasonal variations in external concentrations, but this requires time to acclimate. However, some macroinvertebrates, notably mayflies [1], are particularly sensitive to increases in dissolved ions, and rapid variations in concentration are more stressful compared to slow concentration changes [2].

Anthropogenic salinization also alters ionic composition [3]. The mode of ion toxicity action to aquatic macroinvertebrates is uncertain [4] but appears less related to direct disruption of osmoregulation [2,5], and more related to the actions of specific ions [6,7]. The apparent toxicity of a specific ion, however, is mediated by the presence of other ions [8–10], resulting in wide ranges in effect levels for a given ion in toxicity tests. Additionally, although disruption of osmoregulation may not be a direct cause of mortality,

the energy cost needed to regulate internal ionic balance has been demonstrated to have sublethal effects at concentrations below chronic endpoints [11–13], and consequential at the assemblage level [14].

The natural background concentration and composition of dissolved ions in rivers and streams is largely governed by the geology of the catchment area. In Ohio, streams draining the Western Allegheny Plateau (WAP) tend to average 100–200 SC units lower than streams draining ecoregions in the glaciated portion of the state (Figure 1), and tend to have sulfate as the dominant anion. However, across all ecoregions, agriculture, mining, impervious cover, and wastewater loadings increase the concentrations of dissolved ions. Of these sources, loadings from impervious cover and mining are the most threatening, as those come in either high pulses [15], or steady high doses [16]. Ecoregions comprising the glaciated portion of the state are the Huron-Erie Lake Plain (HELP); the Erie-Ontario Lake Plain (EOLP); the Eastern Corn Belt Plains (ECBP); and the Interior Plateau (IP).

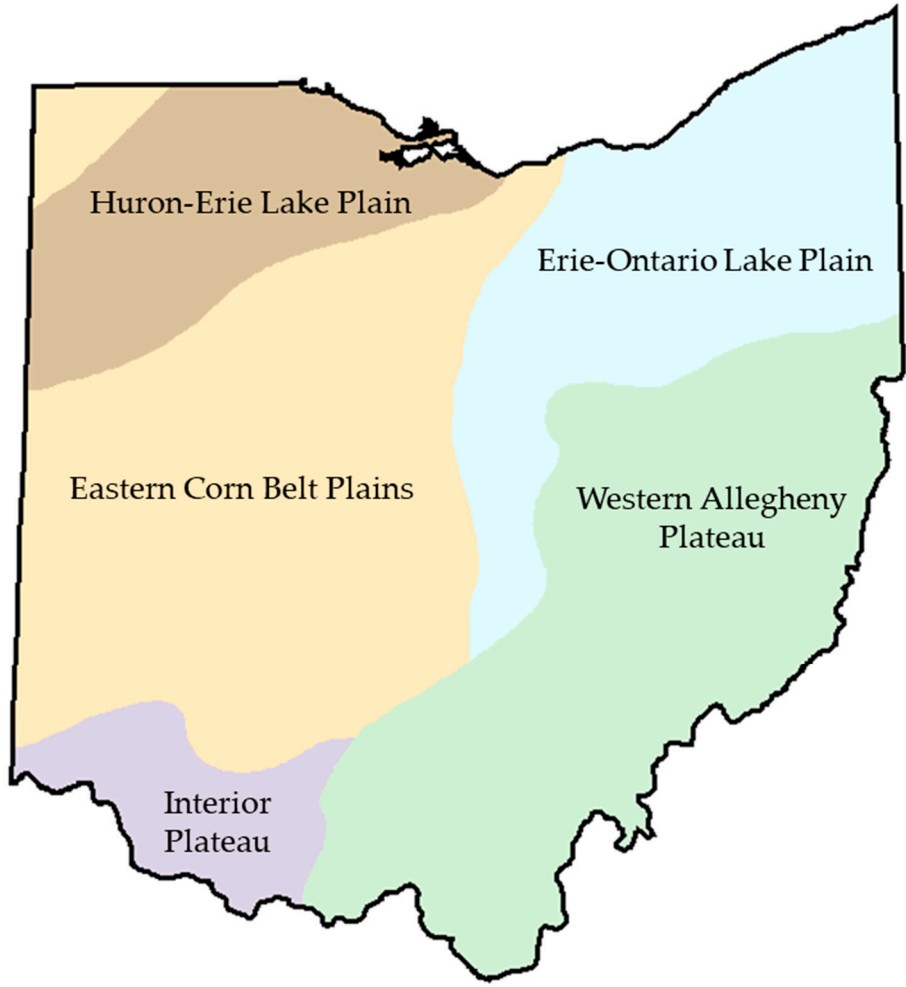

**Figure 1.** Level III ecoregions of Ohio.

Traditional laboratory toxicology tests focusing on lethality, or mesocosm experiments on the toxicity of TDS or specific ions (i.e., chloride or sulfate), have resulted in conditional endpoints. That is, endpoints are conditioned on various factors such as acclimation, ionic composition, synergistic effects with other toxicants (e.g., metals), and secondary effects. Additionally, many laboratory test organisms show efficient osmoregulation, as evidenced by lethal endpoints near isotonic concentrations. In the net, the results from laboratory and mesocosm studies with lethal endpoints can be interpreted as maximum thresholds not to exceed. However, mesocosm studies that account for sublethal [12] and assemblage-

level effects [14] have produced protective endpoints in line with observations from field studies [17], and that are an order of magnitude less than laboratory-derived endpoints.

Species sensitivity distributions (SSDs) can be generated from field collections by observing the occurrence of species or taxa over a specific stressor gradient. For example, a field data set that has matching biological and chemical observations (of chloride, for example) can be arrayed by increasing chloride concentration for each taxon, and the corresponding 95th percentile concentration for each taxon is identified as an extirpation concentration (XC95). Then, the 5th percentile of the XC95s for all taxa is taken as the hazard effect concentration (HC5); a concentration that should be protective for most taxa. This approach can better capture the range and complexity of exposures more typical of a given environmental setting [17]. An example of this method from the data set used in this study is illustrated in Figure 2.

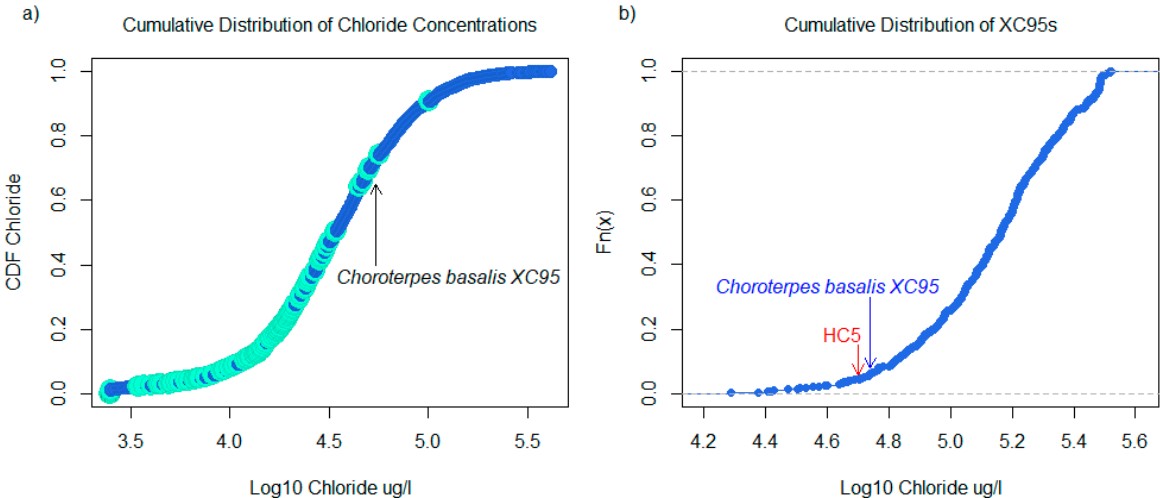

**Figure 2.** (**a**) The cumulative distribution of chloride concentrations from all sites. Site where the mayfly *Choroterpes basalis* occurs are highlighted in aquamarine and with a larger point size. The 95th percentile chloride concentration specific to *C. basalis* is noted as the XC95. (**b**) Cumulative distributions of XC95s for all taxa, with the XC95 for *C. basalis* noted. The hazard concentration (HC5) is taken as the 5th percentile from the distribution of XC95s.

Similarly, biological indicators or multi-metric indexes (MMI) have long been used in direct gradient analysis to identify environmental thresholds for various pollutants including nutrients [18], sediment [19], and metals [20]. MMIs have the advantage of directly expressing whether a waterbody is meeting a specified status (e.g., the goal use of the Clean Water Act), and environmental thresholds identified using MMIs thusly communicate that aspect of risk. However, the need to "decide in a field situation whether a criterion is too high or too low or just right", as suggested by US EPA [21], remains. That decision should necessarily occur in the context of the environmental setting and protection goals for a specific waterbody, a measurement of one or more communities, and a comparison of pollutant concentrations to both existing water quality criteria (typically laboratory-derived) and other empirically derived thresholds. Essentially, what this implies is that a criterion should be contextualized against the biocondition gradient (BCG).

The BCG concept serves as a useful backdrop for evaluating whether existing water quality criteria are potentially under- or over-protective, where other empirically derived thresholds lie along the gradient, and the potential for determining whether concentrations are contributing to use impairment for a given situation. The BCG describes the expected biological condition relative to a gradient of increasing environmental stress by partitioning the response into five (or more) narrative categories as: (1) natural or native condition, (2) minimal changes in structure and function, (3) evident or moderate changes in structure and minimal change in function, (4) major changes in structure and moderate changes in function, and (5) severe changes in structure and loss of function [22]. These narratives

recognize that some changes to assemblage structure can occur without significant loss of ecosystem services.

The strategy used here to identify relevant thresholds for chloride and sulfate used both the SSD approach, and modeling attainment of ecoregion-specific benchmarks for macroinvertebrates using a Bayesian mixed logistic regression model. When logistic regression is applied in this context, the results are typically expressed as the probability of an outcome, in this case attainment of the benchmark, with a credible interval informed by the uncertainty from all covariates included in the model. Plotting the probability of attainment ($\pm$ the credible interval) over the full array of either chloride or sulfate concentrations, and noting where the respective HC5s and existing or proposed laboratory-based water quality criteria fall on the respective continuums, frames how those endpoints might apply to a given situation in the field.

## 2. Materials and Methods

### 2.1. Data Set

Observations of water chemistry and macroinvertebrates were obtained from 4973 unique site-year combinations from 2003 through 2019. All sites drained catchments less than 10,000 mi$^2$. Habitat observations were included at 4634 of those sites. Habitat observations were summarized as Qualitative Habitat Evaluation Index (QHEI) [23,24] scores. The QHEI is a qualitative visual assessment of functional aspects of stream macrohabitat (e.g., substrate quality, amount and type of cover, riparian width, siltation, channel morphology). Water quality samples were collected four to six times during a summer index period of 15 June–15 October. Water quality observations typically include field measurements of dissolved oxygen, temperature, pH and specific conductivity, plus laboratory reported values for various parameters, including metals, anions (e.g., chloride and sulfate), and nutrients (e.g., total phosphorus, nitrate and nitrite nitrogen, ammonia nitrogen, and total Kjeldahl nitrogen). All laboratory reported values are based on standard methods listed in [25,26]. To eliminate highly polluted sites, those with average pH < 6.7 or ammonia nitrogen greater than 0.1 mg/L were excluded from all further analyses. Distributions of selected water chemistry parameters and QHEI scores by ecoregion are shown in Figure 3.

Macroinvertebrate communities at each site were collected following methods described in [27,28] by sampling all available habitat types with net sweeps and by hand, and the resulting information was coded to 1 for presence and 0 for absence. Additionally, if the assemblage at a site met the ecoregion benchmark based on its MMI score [29] it was assigned a score of 1, otherwise 0. The scoring algorithm for the MMI and ecoregion benchmarks are included as Supplemental Material. The MMI has a scale of 0 to 60, and the ecoregional benchmarks reflect differences in expectations driven by prevailing land use and surficial geology. The benchmarks defining a score reflecting Warmwater Habitat (i.e., the base aquatic life use goal) are as follows: HELP—18, IP—23, EOLP and ECPB—26, WAP—34.

### 2.2. Individual Taxa Responses

Taxa responses over gradients of chloride or sulfate were modeled in two ways. The first using logistic regression, and the second as species sensitivity distributions. For both methods, chemistry parameters were log10 transformed and averaged by site. For the logistic regression models, only taxa with a minimum of 10 observations for either chloride or sulfate were modeled, and only models with positive or negative slopes significant at the $p < 0.01$ level were retained. Note that the $p$-value in this context is arbitrary but was chosen to yield a reasonable number of responses for subsequent plots. Individual taxa responses were expressed as probabilities at the 5th, 10th, 25th, median, 75th, 90th and 95th quantiles of chloride and sulfate, and plotted as distributions at those respective quantiles. All statistical models were run in R 4.0.3 [30].

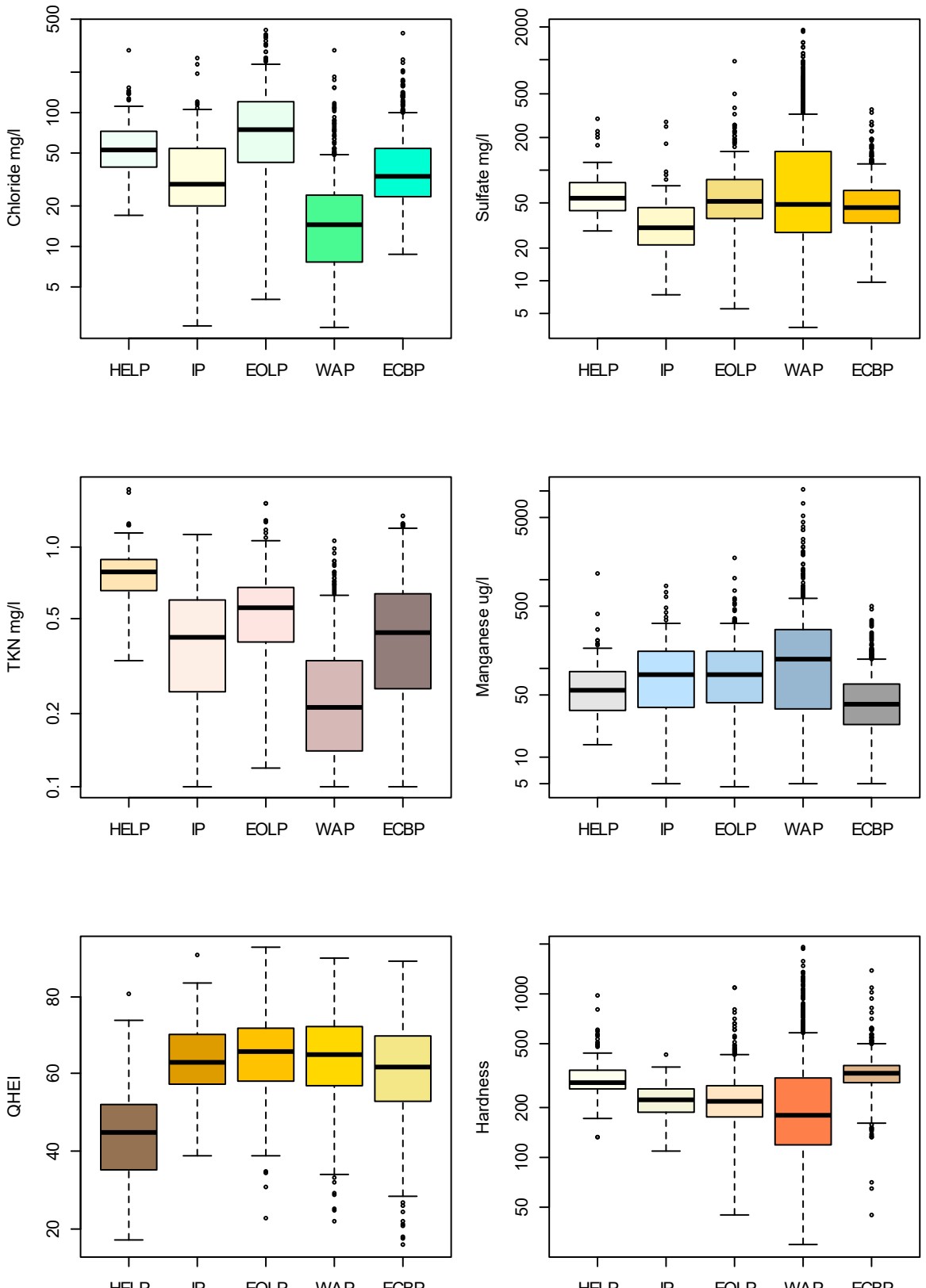

**Figure 3.** Distributions of water quality parameters and QHEI scores by ecoregion in Ohio. Ecoregion acronyms are as follows: HELP—Huron-Erie Lake Plain; IP—Interior Plateau; EOLP—Erie-Ontario Lake Plain; WAP—Western Allegheny Plateau; ECBP—Eastern Corn Belt Plains. Note that the y-axes for the water chemistry constituents are log10-scaled.

### 2.3. Species Sensitivity Distributions

SSDs were derived from field data following the methods of Cormier and Suter [31], and were based on taxa with observations for either chloride or sulfate from at least 10 different sites. From a starting pool of 981 taxa, 665 taxa had at least 10 observations, and of those, 578 had companion observations of chloride, and 556 had companion observations for sulfate. For chloride, the statewide data set was used. For sulfate, SSDs were constructed from the statewide data, and data exclusive to the WAP; the latter because sulfate is the dominant anion in the WAP. Log transformed chloride and sulfate values averaged by site were used in the calculations. The respective HC5s were obtained from the collection of XC95s resulting from the SSDs.

### 2.4. Bayesian Mixed Logistic Regression Model

A Bayesian mixed logistic regression model was developed for headwater streams (<20 mi$^2$) to assess the influence of chloride and sulfate on observing macroinvertebrate scores meeting their expected benchmark. Headwaters are modeled as they are most directly impacted by road salt and mining legacies. This model included several other variables that represent common stressors. Specifically, total Kjeldahl nitrogen (TKN) to represent organic and nutrient enrichment, manganese, and the QHEI as an indicator of habitat quality. Manganese is associated with mining, low dissolved oxygen, and streams draining soils with low redox potential. Note that because of a high degree of collinearity between chloride and TKN (r = 0.60 across ecoregions), especially in the WAP and ECBP (See Appendix B), the residuals from a regression of TKN on chloride were used in lieu of TKN, and are hereafter referred to as TKNr. This approach isolates a TKN effect divorced from chloride, but not vice versa. Chloride may facilitate enrichment through desorption of sediment bound phosphorus and the release of organic nitrogen from sediments [32], and chloride and recalcitrant organic nitrogen are discharged from wastewater plants. Thus, a singular chloride effect cannot be completely isolated, especially in the WAP and ECBP. Note that only records with complete cases for all covariates were included. In this model, stressors were considered fixed effects and ecoregion a random effect. Thus, the model is stated as:

$$Y_{ir} \sim Bernoulli(\pi_{ir}) \tag{1}$$

$$logit(\pi_{ir}) \sim x'_{ir}\beta + \gamma_r \tag{2}$$

$$\gamma \sim Normal\left(\mu, \sigma^2\right) \tag{3}$$

where $Y_{ir}$ is the observation of whether a macroinvertebrate assemblage at a given site within an ecoregion is meeting its benchmark, $\pi_{ir}$ is a vector of probabilities, $x_{ir}$ is a vector of covariates, $\beta$ is a vector of coefficients, and $\gamma$ is one of five ecoregions denoted by the subscript $r$.

The model was compiled in JAGS 4.3.0 using the rjags 4.0.4 package, run using the autorun.jags function in runjags 2.2.0–2, and assessed using the coda 4.0.4 package. Three Markov chains with a burn-in of 51,000 and 155,842 samples were assessed visually for convergence in trace plots and using the Gelman-Rubin statistic (at ≤1.01 to indicate convergence). The Gelman-Rubin statistic essentially tests whether regression parameters estimated from the separate chains follow similar distributions. A sample length of 155,842 suggested by the Raftery and Lewis [33] diagnostic resulted in an effective sample size of at least 450 for the estimated model coefficients. The Raftery and Lewis diagnostic is used to control the error surrounding regression parameter estimates. The chains were initialized with a range of starting values suggested by standard logistic regression. The model code is supplied as Appendix A.

Inferences from the model were gathered by sampling the posterior distribution using the BayesPostEst 4.0.5 package. Predicted probabilities of meeting the macroinvertebrate benchmarks for MMI scores within an ecoregion as a function of either chloride or sulfate were obtained based on ecoregion median values for other predictors in the model. Simi-

larly, the relative influence of each parameter in the model was assessed by compiling first differences [34]. Here, first differences estimate the change in probability of meeting the macroinvertebrate benchmark when a given predictor moves across the middle quartile of its within-ecoregion range while holding other predictors at their medians. Model fit for each ecoregion was assessed by obtaining the area under the Receiver Operating Characteristic (ROC) curve, and the Hosmer-Lemeshow [35] Ĉ statistic.

## 3. Results

### 3.1. Taxa Sensitivities

Distributions of individual taxa responses modeled by logistic regression are plotted for selected quantiles of chloride and sulfate in Figure 4. The distributions are stratified by whether taxa showed a decrease along the gradient, or an increase. Taxa sensitive to chloride appear to decrease across the range of chloride concentrations. For sulfate-sensitive taxa, a decreasing trend across the gradient is not apparent until concentrations exceed the 50th percentile. The HC5 for chloride is 52 mg/L and that corresponds to the 68th percentile concentration in the statewide data. The CCC of 230 mg/L published by US EPA [36] corresponds to the 99th percentile concentration. Similarly, the HC5 concentration for sulfate from the statewide set is 90 mg/L and corresponds to the 76th percentile. When calculated from data from the WAP, the HC5 concentration is 152 mg/L, and that corresponds to the 72nd percentile concentration for the WAP. The biggest apparent disparity between increasing and decreasing taxa appears over low chloride concentrations, and likely reflects the sensitivity of certain mayfly taxa to ions. Mayfly taxa had a higher-than-expected frequency in the lower 5th percentile of the XC95 for chloride relative to other taxa groups ($\chi^2$ = 19.093, df = 6, $p$-value = 0.004009). Of the 29 taxa comprising the lower 5th percentile, 10 were mayfly (3.26 expected; see Appendix C for observed and expected frequencies by major taxa group). Taxa groups in the lower 5th percentile of the XC95s from the SSDs constructed for the statewide sulfate data were marginally similar ($\chi^2$ = 11.5240, df = 6, $p$-value = 0.07348), but those for the WAP also had a higher than expected incidence of mayfly taxa in the lower 5th percentile ($\chi^2$ = 20.6040, df = 6, $p$-value = 0.00216; 2.5 taxa expected, 7 taxa observed).

### 3.2. Bayesian Logistic Regression Model

Estimates of posterior means, the Gelman-Rubin convergence diagnostics and effective sample sizes are shown in Table 1. Model fits were generally good, as suggested by the Hosmer-Lameshow test (Table 2), except for the WAP ecoregion, where a subsequent investigation revealed a nonlinear relationship between macroinvertebrate index scores and sulfate. At relatively low concentrations of sulfate (i.e., for the WAP < ~50 mg/L), index scores show a slight increase over increasing sulfate concentrations. Sulfate and hardness are strongly linearly related in the WAP ($r^2$ = 0.74), and the tendency for increasing index scores may reflect an ameliorative effect from hardness [7,37,38] on ion toxicity in general, at least over a range of relatively low hardness concentrations (<160 mg/L). Nevertheless, the AUC score for the WAP indicated good classification accuracy. Area Under the Curve (AUC; i.e., the area under ROC curve) scores for the IP, EOLP and ECBP indicated fair accuracy, while the score for the HELP suggested poor accuracy. Because the water chemistry parameters were measured over roughly similar scales in log space, their coefficients can be compared directly to infer relative effect size. This suggests that organic and nutrient enrichment isolated from the additional effect of chloride, as represented by TKNr, is relatively more consequential in general than the other water quality measures, and comports with findings for Ohio [39]. However, first differences (Figure 5) provide a more direct comparison of effect sizes within ecoregion, and suggest that for the IP and EOLP, where TKN and chloride are not strongly correlated, the effect size for chloride is appreciable and similar to the WAP and ECBP. Sulfate and manganese have the most pronounced effect in the WAP. Note that manganese and sulfate concentrations in the WAP are not strongly correlated (r = 0.13). Because first differences are computed over

the interquartile range (IQR) for a given variable while holding other variables at their medians, the magnitude of the IQR will partially dictate the size of the first difference. Plotting the first differences on scaled IQRs for a given parameter helps to place the differences into context. For example, the first differences for TKNr in the ECBP and IP were similar, but the IQR in the ECBP was narrower, suggesting a relatively larger effect per unit change in the ECBP. Similarly, the WAP should be the most sensitive of the ecoregions to a change in chloride concentration. The conspicuously muted response to chloride and organic enrichment in the HELP shown in Figure 5 reflects both the pervasively degraded habitat, and the narrow interquartile (IQR) range of TKN within the ecoregion (Figure 3). Axiomatically, a change in habitat quality in the HELP is likely to have a bigger effect than a change in any one water quality parameter.

**Table 1.** Parameter estimates and diagnostic statistics for the Bayesian logistic regression model.

| | Posterior Means and (SD) [95% Credible Interval] | Gelman-Rubin Point and Upper C.I. | Effective Size |
|---|---|---|---|
| Intercept | 0.00 (0.010) [−0.19; 0.20] | 1.00–1.00 | 29,242 |
| Chloride | −2.00 * (0.189) [−2.37; −1.63] | 1.01–1.03 | 1034 |
| QHEI | 0.04 * (0.005) [0.03; 0.05] | 1.00–1.00 | 6816 |
| rTKN | −3.15 * (0.315) [−3.77; −2.53] | 1.00–1.00 | 10,256 |
| Mn | −0.85 * (0.141) [−1.13; −0.59] | 1.00–1.00 | 1423 |
| SO4 | −1.09 * (0.159) [−1.40; −0.78] | 1.00–1.01 | 1354 |
| HELP | 16.83 * (1.375) [14.19; 19.62] | 1.01–1.01 | 484 |
| IP | 16.70 * (1.364) [14.09; 19.47] | 1.01–1.01 | 465 |
| EOLP | 16.53 * (1.397) [13.84; 19.37] | 1.01–1.01 | 451 |
| WAP | 15.45 * (1.342) [12.89; 18.18] | 1.01–1.01 | 465 |
| ECBP | 15.44 * (1.322) [12.90; 18.13] | 1.01–1.01 | 463 |

* 0 outside 95% credible interval.

**Table 2.** Model fit statistics.

| | HL $\chi^2$ | P | AUC |
|---|---|---|---|
| HELP | 5.31 | 0.72 | 61.8 |
| IP | 9.49 | 0.30 | 78.3 |
| EOLP | 3.72 | 0.88 | 76.1 |
| WAP | 18.92 | 0.02 | 83.6 |
| ECBP | 6.91 | 0.55 | 70.1 |
| Over-all | 9.89 | 0.27 | |

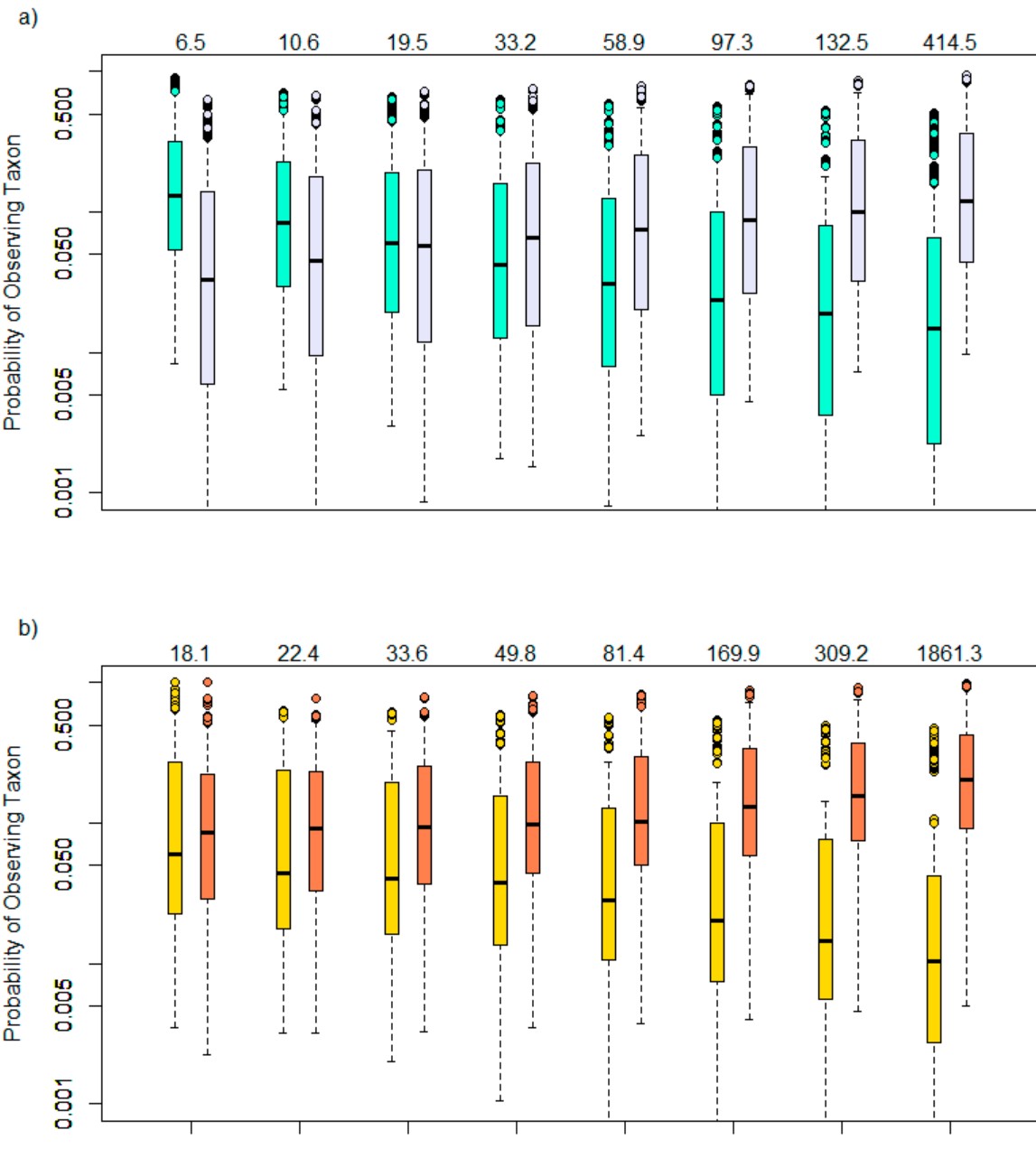

**Figure 4.** Distributions of taxa observation probabilities (y-axes) from logistic regression plotted by quantile intervals (e.g., <5th, 5th–10th, etc.); concentrations for the stated quantiles are arrayed across the top margins. (**a**) Probabilities for chloride where taxa having decreasing slopes are shown in light green, and those with increasing slopes in light purple. For reference, the HC5 from the species sensitivity distribution constructed from statewide data is 52 mg/L. The chronic continuous concentration (CCC) published by US EPA [36] is 230 mg/L. (**b**) Probabilities for sulfate where taxa having decreasing slopes are shown in yellow and those with increasing slopes are shown in orange. For reference, the HC5 from the species sensitivity distribution constructed from the WAP subset is 152 mg/L, and the HC5 from the statewide SSD was 94 mg/L. The Criterion Maximum Concentration (CMC) from the Illinois toxicity model [37] would default in many cases to 500 mg/L for Western Allegheny Plateau headwaters because of high levels of hardness associated with mine drainage. Note that acute to chronic ratios are typically ~10 [40].

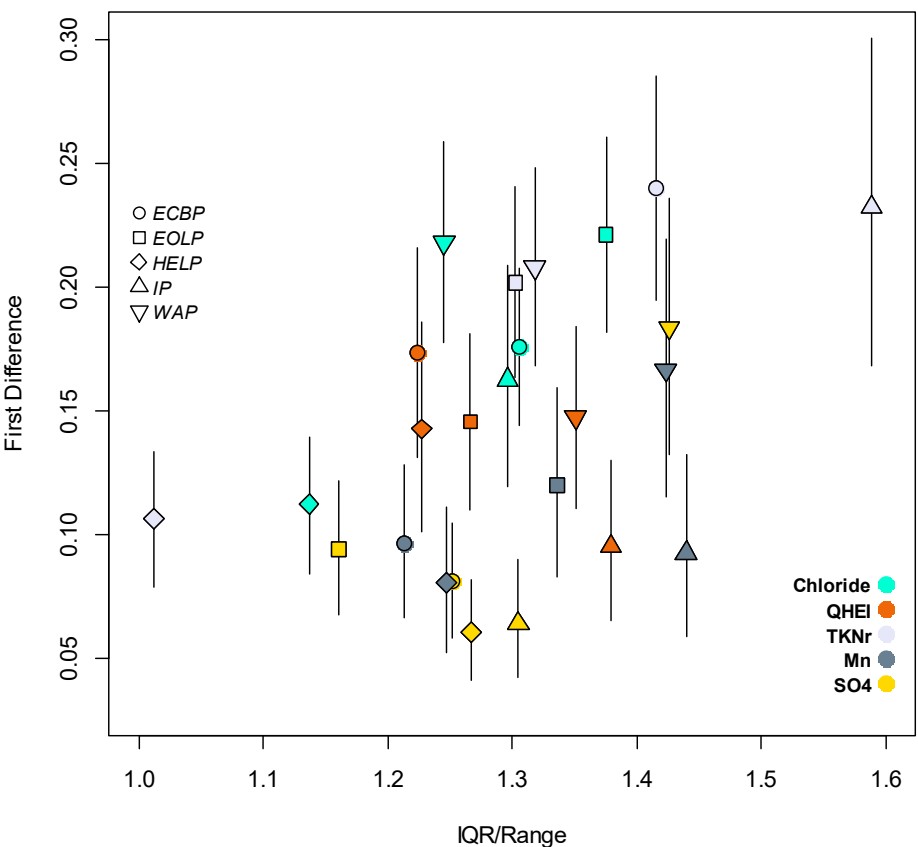

**Figure 5.** First differences (+/− 95% credible interval) as a function of scaled interquartile ranges. Note that absolute values for the first differences are plotted on the y-axis.

Figures 6 and 7, respectively, show the predicted probabilities of meeting ecoregion-specific macroinvertebrate MMI benchmarks as a function of chloride and sulfate concentrations when the other covariates included in the model are held at their ecoregion median values. Median effect levels (i.e., the point where the probability of meeting the ecoregion benchmark is 0.5) for all the ecoregions are less than the chloride CCC of 230 mg/L. Only in the IP does the upper credible interval suggest a better than even chance of meeting the benchmark if the CCC is exceeded, but the CI at that point encompasses a wide range. Relative to the HC5 (52 mg/L), the median effect level was lower in the HELP, WAP and ECBP, and higher in the IP and EOLP. As was evident with first differences, the lower effect level in the HELP reflects the magnitude of poor habitat quality. For the WAP and ECBP, the lower effect levels are likely due, in part, to the added contribution of TKN, and in the WAP, higher levels of sulfate. In the IP and EOLP where the effect of chloride is less entangled with TKN, the median effect levels may approximate an operational benchmark in lieu of the CCC.

For sulfate, median effect levels in the EOLP, WAP and ECBP approximate the respective HC5s, especially in light of where the CIs intersect that point on the graph. The comparatively high median effect level in the IP reflects relatively low chloride concentrations. Similarly, lower median chloride and TKN concentrations in the WAP, relative to the other ecoregions, also results in a higher median effect level. If the median value for chloride from the EOLP is substituted in the IP, then the median effect level for the IP becomes 70 mg/L, as might be expected given fixed effects and similar intercepts between the two ecoregions. The wide credible intervals for the IP likely reflect uncertainty imposed by collinearity between sulfate and chloride (see Appendix B for correlations by ecoregion). Collectively, these results suggest that sulfate concentrations approaching 100 mg/L are

consequential, and concentrations exceeding 300 mg/L are generally incompatible with macroinvertebrate communities meeting their benchmarks.

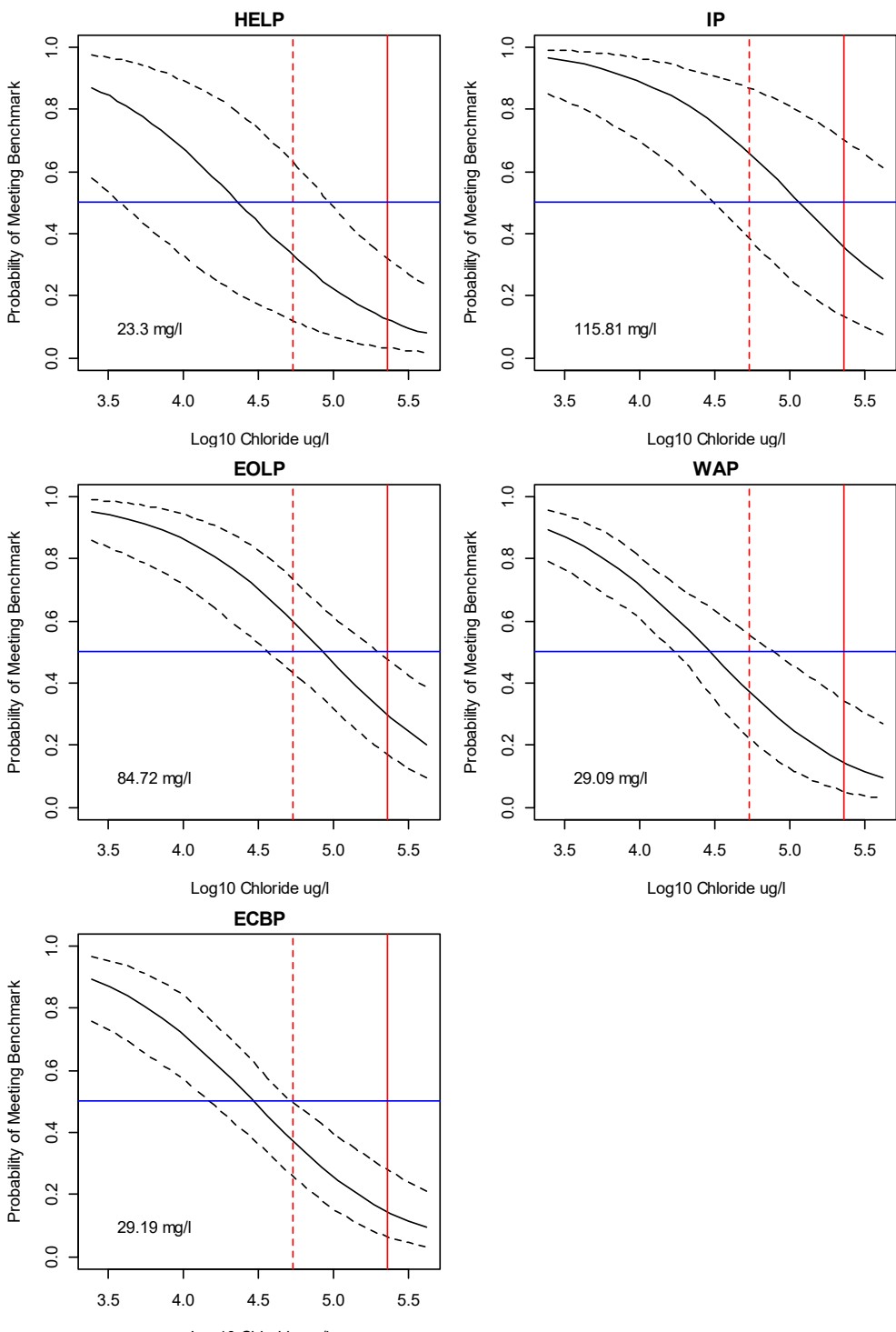

**Figure 6.** Predicted probabilities and 95% credible intervals of macroinvertebrate communities meeting their ecoregion benchmark as function of chloride concentrations when other model parameters are held at their ecoregion medians. The vertical dashed red line is the HC5 (52 mg/L), the solid vertical red line is the CCC (230 mg/L). The blue horizontal line is drawn at *p* = 0.5 for reference. Respective median effect levels in standard units are inset as text in the lower left of each plot.

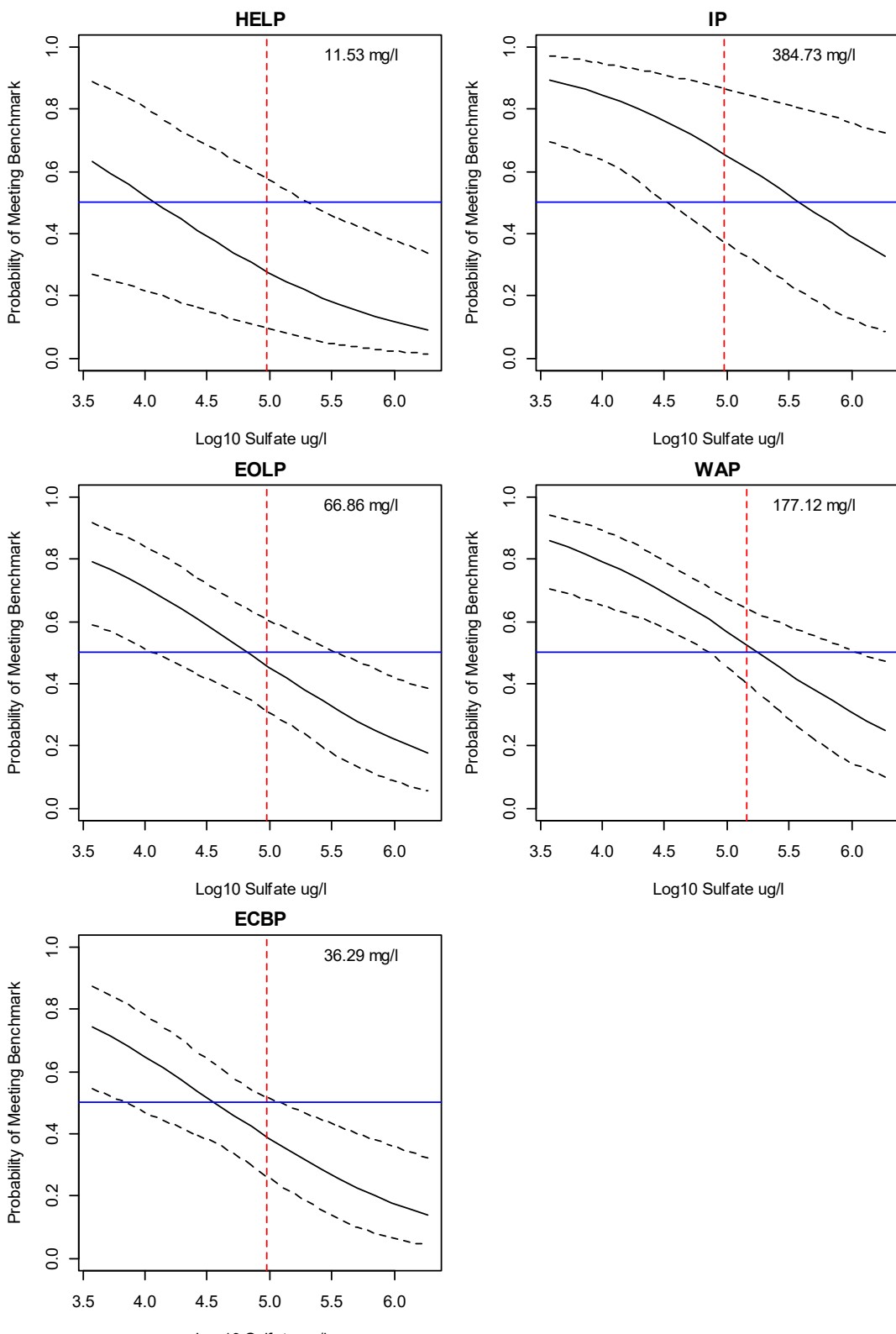

**Figure 7.** Predicted probabilities and 95% credible intervals of macroinvertebrate communities meeting their ecoregion benchmark as function of sulfate concentrations when other model parameters are held at their ecoregion medians. The vertical dashed red line is the HC5, equating to 90 mg/L for the four ecoregions exclusive of the WAP, and 152 mg/L for the WAP. The blue horizontal line is drawn at $p = 0.5$ for reference. Respective median effect levels in standard units are inset as text in the upper right of each plot.

## 4. Discussion

The individual taxa responses from the logistic regression models, when considered collectively and, respectively, against chloride and sulfate (Figure 4), broadly reflect expectations suggested by the present state of knowledge. Mayfly taxa have a relatively high frequency in the lower 5th percentile of the XC95s for chloride and sulfate, but some mayfly taxa are tolerant of chloride, as evidenced by increasing slopes in the logistic regression models (Figure 8a). This generally comports with findings from other studies e.g., [3,4,41,42], and the observation that some mayfly taxa can acclimate to increased salinity by reducing the number of chloride cells on tracheal gills in response to increased salinity [43]. Interestingly, of the 166 taxa showing a directional response to chloride, 75 (45%) have an increasing response, and of those, 21 taxa are considered sensitive [28], at least to other types of pollution. This suggests that turnover of taxa can maintain richness components of MMI scores consistent with attaining benchmarks. However, Drover et al. [44] cautioned that species turnover could impact functional aspects of an assemblage in a negative way that is not capture by richness metrics, implying that when evaluating biological impacts associated with chloride, functional differences associated with species turnover should be considered. Moreover, in instances where measured chloride concentrations are low, the presence of chloride tolerant taxa may signal a history of exposer [45].

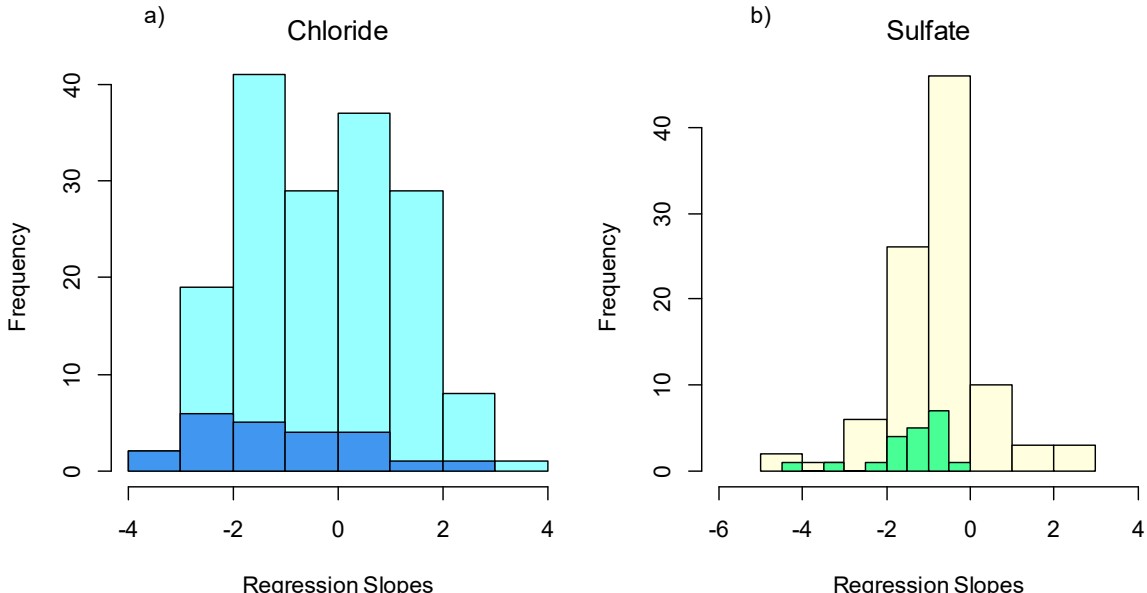

**Figure 8.** Histograms of slopes (x-axis) for individual taxa reported from logistic regression for (**a**) chloride and (**b**) sulfate. The subset of slopes for mayfly taxa are shown as inset overlaying histograms.

Most of the taxa showing a directional response against sulfate had decreasing responses, and of those with increasing responses, none was from mayfly taxa (Figure 8b). This suggests a lower overall ability for macroinvertebrates to acclimate to higher concentrations of sulfates, and the broader decreasing response across taxa may reflect a more direct toxic response to the associated cation. Magnesium is more toxic than either sulfate or chloride [6,7], and has a more direct effect due to the activity of the cation [46].

### Application of Results to Management

The distributions shown in Figure 4 highlight potential shifts in assemblage structure that occur over each gradient. With respect to chloride, the HC5 (52 mg/L) from the statewide data coincides with a point on the gradient (i.e., the 68th percentile) where assemblage structure is materially altered, as inferred by the increasing disparity between chloride-tolerant and chloride-sensitive taxa past that point. In terms of the BCG, this

might coincide with moderate changes in structure and minimal to moderate changes in function. The HC5 for sulfate from the WAP SSDs (152 mg/L) is similarly situated in terms of the percentile concentration, but a relative increase in sulfate-tolerant taxa is not as pronounced as for chloride. The probabilities plotted in Figures 6 and 7 show where changes in function are increasingly likely to have a material consequence, specifically, loss of the beneficial use when macroinvertebrates communities fail to meet their benchmarks. The CCC of 230 mg/L for chloride is clearly past the point on the BCG where communities are likely to meet their benchmarks. Based on hardness values typical for WAP headwaters affected by mine drainage, the CMC value for sulfate from the Illinois laboratory model defaults to 500 mg/L in most cases, and represents a point on the BCG where the risk of losing function is high, given the probability of meeting the WWH benchmark at that point is ~0.38 (when holding all other parameters at their respective medians). An acute to chronic ratio of 10 [40] gives a CCC of 50 mg/L, but that appears overly protective. A study by Wang et al. [47] found acute to chronic ratios of ~3.33 for three invertebrates tested against sodium sulfate. Cast in that light, a CCC of ~160 mg/L appears to match observations here.

In terms of managing condition status, the HC5s best represent either protective caps for existing high-quality waters, or restoration targets for presently impaired waters in some cases, but not as criteria to be applied independently. Coming from the other direction, and with respect to chloride, waters with average chloride concentrations that exceed the CCC of 230 mg/L (over a suitable averaging period) should be considered impaired. For waters with biological impairment linked to chloride, but having measured concentrations less than the CCC, a site or waterbody-specific restoration target should be developed and informed by the Bayesian logistic regression model.

Management decisions can be informed by the Bayesian model by examining first differences and effects over observed cases [48]. Figure 9 plots expected effects associated with four levels of chloride given values for the four other covariates within three slices of the observed data. The examples are drawn from the Eastern Corn Belt Plains (ECBP) and Erie-Ontario Lake Plains (EOLP). The levels for chloride are given by sequencing the ecoregion-specific range from the 25th to 90th percentile, and the three data groups correspond to the lower, middle and upper 25th quantiles of TKN for the respective ecoregions (n.b., the TKNr residuals associated with those strata are used in the calculation). In the ECBP, a reduction in TKN from the upper quartile to middle quartile appears to have a substantial effect, whereas in the EOLP, the effect is less dramatic. Because these effects are modeled on observed data, distributions of other covariates in the data slices will influence the outcome. In the ECBP, streams with high TKN also tend to have poor habitat (mean QHEI = 53 ± 15) compared to streams with TKN in the middle quartile (QHEI = 63 ± 11). In the EOLP, the distributions of QHEI scores are similar between the upper and middle quartile slices (64 ± 11 and 66 ± 11, respectively). Both examples illustrate the need to consider the influence of multiple stessors in restoration efforts, especially in the context of pollution from diffuse sources. In the case of the ECBP, restoration efforts that address both nutrient enrichment and habitat quality are likely to be more successful than a singular effort. Failure to observe multiple stressors has resulted in limited success in restoration efforts [49,50], and it is increasingly recognized that the biological response to habitat restoration is likely to be tempered by watershed-scale characteristics [51–54]. In the EOLP, where salinization from deicers is a significant problem, the success of best management practices (BMPs) directed at reducing chlorides, when measured by aquatic life, will depend on levels of other stressors. In other words, the need for BMPs in case of deicers is singular, but assessment of efficacy is necessarily multivariate, especially in light of the potential range of water quality and biological effects associated with chloride enrichment [32,55]. Similar plots for the HELP and IP ecoregions are provided in Appendix D.

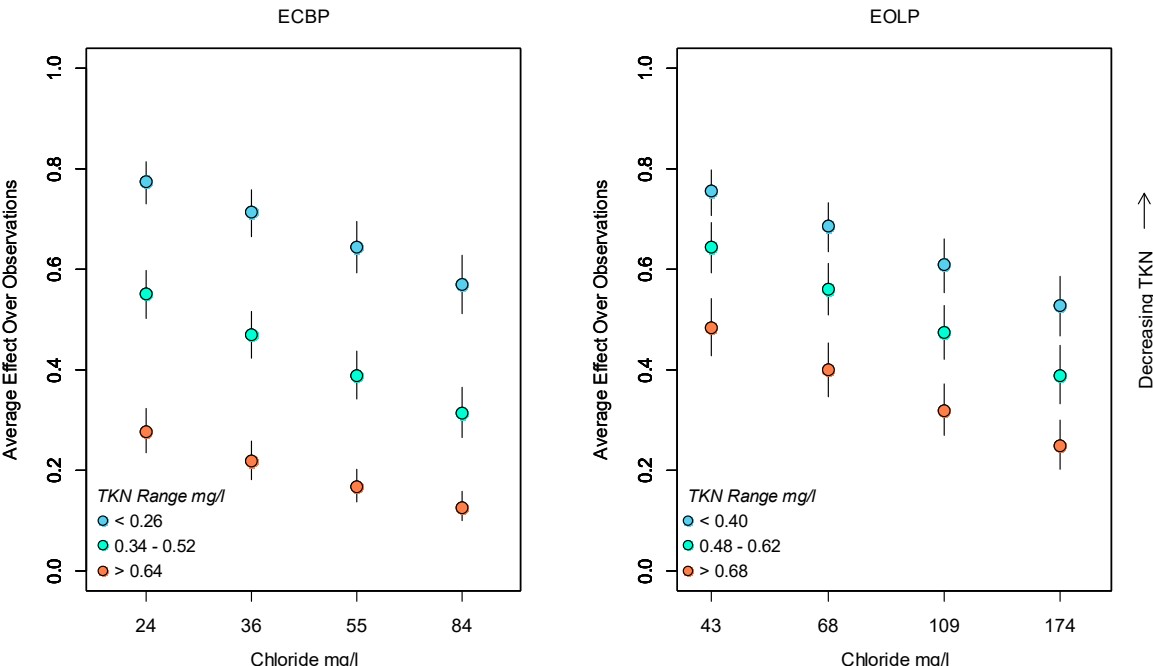

**Figure 9.** Probabilities of observing macroinvertebrate communities meeting their ecoregion benchmark given relevant levels of chloride and observations of covariates within three data slices defined by quantile levels of TKN (lower, middle and upper 25th quantiles). The chloride levels were obtained by sequencing the range from the 25th to 90th percentiles (in log space). Examples are for the Eastern Corn Belt Plains (ECBP) and Erie-Ontario Lake Plain (EOLP).

In the Wester Allegheny Plateau (WAP), where ions from active and legacy mining are a concern, the model suggests that management and reclamation directed toward reducing manganese concentrations are likely to have a significant effect across a wide range of sulfate levels, especially if manganese concentrations can be reduced to less than ~35 ug/L (Figure 10). Passive treatment systems for manganese are capable of high removal efficiencies and are relatively inexpensive [56–58], whereas removing sulfate is likely impractical. The results for the WAP also illustrate the potential difficulty in defining and administering parameter-specific water quality criteria when the effects of pollutants are additive. Laboratory studies that identify the mode of action and hazard concentrations are necessary to establish a basis for causation for a given parameter, but should be complemented by effect levels and models derived from field studies. Both should inform causal determinations, listing decisions, and restoration targets for impaired waters.

## 5. Conclusions

Water quality standards for the protection of aquatic life are intended to maintain attainment of the beneficial use. Compliance with a given standard has traditionally been judged by comparing water quality observations against a numeric endpoint derived from laboratory tests. However, the fact that changes in biological communities are detectable across entire pollution gradients, as was observed here for chloride and sulfate, calls for a more holistic framework for managing pollution. This framework should include protective endpoints derived from field and laboratory studies; the former to identify thresholds necessary to maintain existing and relatively unperturbed conditions, and the latter as backstops against overt pollution. The results here clearly demonstrate that the existing CCC for chloride (230 mg/L) is under-protective as a backstop, and that HC5s for chloride (52 mg/L) and sulfate (152 mg/L for the WAP and 90 mg/L for all other ecoregions) are needed to maintain a relatively unperturbed condition. Biological communities should be evaluated against position on the pollution gradient, and where impairment is observed, restoration strategies and targets should be informed by all relevant chemical and physical information.

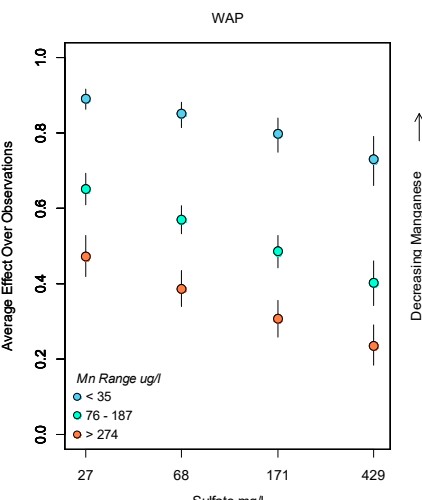

**Figure 10.** Probabilities of observing macroinvertebrate communities meeting the ecoregion benchmark for the Western Allegheny Plateau given four levels of sulfate and observations of covariates within three data slices defined by quantile levels of manganese (lower, middle and upper 25th quantiles). The sulfate levels were obtained by sequencing the range from the 25th to 90th percentiles (in log space).

**Supplementary Materials:** The following are available at https://www.mdpi.com/article/10.3390/w13131815/s1, A list of taxa and extirpation concentrations (XC95) for chloride and sulfate, and development and scoring methods for a macroinvertebrate biotic index based on presence/absence data.

**Funding:** This research received no external funding.

**Institutional Review Board Statement:** Not applicable.

**Informed Consent Statement:** Not applicable.

**Data Availability Statement:** The data presented in this study are available on request from the corresponding author. The data are not publicly available due to agency policy.

**Acknowledgments:** The author thanks the dedicated field staff at Ohio EPA and two anonymous reviewers who provided helpful comments on earlier versions of this manuscript.

**Conflicts of Interest:** The author declare no conflict of interest.

## Appendix A. Model Code for Bayesian Logistic Regression

```
sink("lrmod.txt")
cat("model
{
for(i in 1:N){
ac[i]~dbern(p[i])
logit(p[i])<-alpha + beta1 * cl[i] + beta2 * qhei[i] + beta3 * tkn[i] + beta4 * mn[i] + beta5 *
so4[i] + u[REG[i]]
}
alpha ~ dnorm(0, 100)
sigma_a ~ dunif(0, 100)
tau_a <- 1 / (sigma_a * sigma_a)
for(j in 1:M){
u[j]~dnorm(0,tau_a)
}
beta1 ~ dnorm(0.0,0.00001)
beta2 ~ dnorm(0.0,0.00001)
beta3 ~ dnorm(0.0,0.00001)
```

```
beta4 ~ dnorm(0.0,0.00001)
beta5 ~ dnorm(0.0,0.00001)
}
", fill = TRUE)
sink()
line_init<-list(
list(alpha = 40,beta1 = −2,beta2 = 0.05,beta3 = −2.8,beta4 = −0.5,beta5 = −1.1),
list(alpha = 1,beta1 = −0.1,beta2 = 1,beta3 = −10,beta4 = −10,beta5 = −10),
list(alpha = 0,beta1 = −0.5,beta2 = 0.01,beta3 = −1,beta4 = −1,beta5 = −0.5)
)

modout<-autorun.jags(model = "lrmod.txt",monitor = c("alpha", "beta1","beta2","beta3",
"beta4","beta5","u"),
data = hwdata, n.chains = 3, inits = line_init, startsample = 51,000)
```

**Appendix B. Correlations Between Water Chemistry Measures and Habitat Quality Scores Stratified by Ecoregion. Cl—Chloride; TKN—Total KJELDAHL Nitrogen; Mn—Manganese; SO4—Sulfate; Hard—Hardness**

Huron-Erie Lake Plain
Cl TKN Mn SO4 Hard QHEI
Cl 1.00 0.04 −0.09 0.23 0.21 0.10
TKN 0.04 1.00 0.28 −0.14 0.04 −0.06
Mn −0.09 0.28 1.00 −0.05 0.21 0.10
SO4 0.23 −0.14 −0.05 1.00 0.51 0.12
Hard 0.21 0.04 0.21 0.51 1.00 0.09
QHEI 0.10 −0.06 0.10 0.12 0.09 1.00

———————————————————————

Interior Plateau
Cl TKN Mn SO4 Hard QHEI
Cl 1.00 0.17 −0.24 0.65 0.31 −0.16
TKN 0.17 1.00 0.39 −0.07 −0.53 −0.24
Mn −0.24 0.39 1.00 −0.42 −0.32 −0.22
SO4 0.65 −0.07 −0.42 1.00 0.54 −0.07
Hard 0.31 −0.53 −0.32 0.54 1.00 0.12
QHEI −0.16 −0.24 −0.22 −0.07 0.12 1.00

———————————————————————

Erie-Ontario Lake Plain
Cl TKN Mn SO4 Hard QHEI
Cl 1.00 0.27 −0.48 0.29 0.21 0.06
TKN 0.27 1.00 0.07 0.02 −0.13 −0.11
Mn −0.48 0.07 1.00 −0.10 0.02 −0.23
SO4 0.29 0.02 −0.10 1.00 0.83 −0.09
Hard 0.21 −0.13 0.02 0.83 1.00 −0.08
QHEI 0.06 −0.11 −0.23 −0.09 −0.08 1.00

———————————————————————

Western Allegheny Plateau
Cl TKN Mn SO4 Hard QHEI
Cl 1.00 0.42 0.10 0.24 0.32 −0.05
TKN 0.42 1.00 0.46 0.14 0.23 −0.28
Mn 0.10 0.46 1.00 0.13 0.16 −0.24
SO4 0.24 0.14 0.13 1.00 0.85 0.01
Hard 0.32 0.23 0.16 0.85 1.00 0.01
QHEI −0.05 −0.28 −0.24 0.01 0.01 1.00

———————————————————————

Eastern Corn Belt Plains

Cl TKN Mn SO4 Hard QHEI
Cl 1.00 0.51 0.14 0.26 −0.10 −0.10
TKN 0.51 1.00 0.43 0.21 −0.26 −0.31
Mn 0.14 0.43 1.00 0.29 −0.04 −0.36
SO4 0.26 0.21 0.29 1.00 0.30 −0.25
Hard −0.10 −0.26 −0.04 0.30 1.00 0.00
QHEI −0.10 −0.31 −0.36 −0.25 0.00 1.00

**Appendix C. Frequencies of Taxa Groups Occurring at Less Than or Greater Than, the Hazard Concentration (HC5) for Chloride and Sulfate. Taxa Groups Are: C—Caddisfly, D—Dipterans, M—Mayfly, N—Non-Insects, O—Other, S—Stonefly, T—Midges in the Tribe Tanytarsini. Expected Frequencies are Based on the Formula for χ-Square**

| | Observed | | Expected | |
|---|---|---|---|---|
| Taxa Group | ≤HC5 | >HC5 | ≤HC5 | >HC% |
| Chloride | | | | |
| C | 3 | 67 | 3.5121107 | 66.48789 |
| D | 7 | 176 | 9.1816609 | 173.8183 |
| M | 10 | 55 | 3.2612457 | 61.73875 |
| N | 3 | 108 | 5.5692042 | 105.4308 |
| O | 3 | 105 | 5.4186851 | 102.5813 |
| S | 2 | 16 | 0.9031142 | 17.09689 |
| T | 1 | 22 | 1.1539792 | 21.84602 |
| Sulfate | | | | |
| C | 1 | 69 | 3.5251799 | 66.47482 |
| D | 11 | 163 | 8.7625899 | 165.2374 |
| M | 5 | 58 | 3.1726619 | 59.82734 |
| N | 9 | 94 | 5.1870504 | 97.81295 |
| O | 1 | 105 | 5.3381295 | 100.6619 |
| S | 1 | 16 | 0.8561151 | 16.14388 |
| T | 0 | 23 | 1.1582734 | 21.84173 |

**Appendix D**

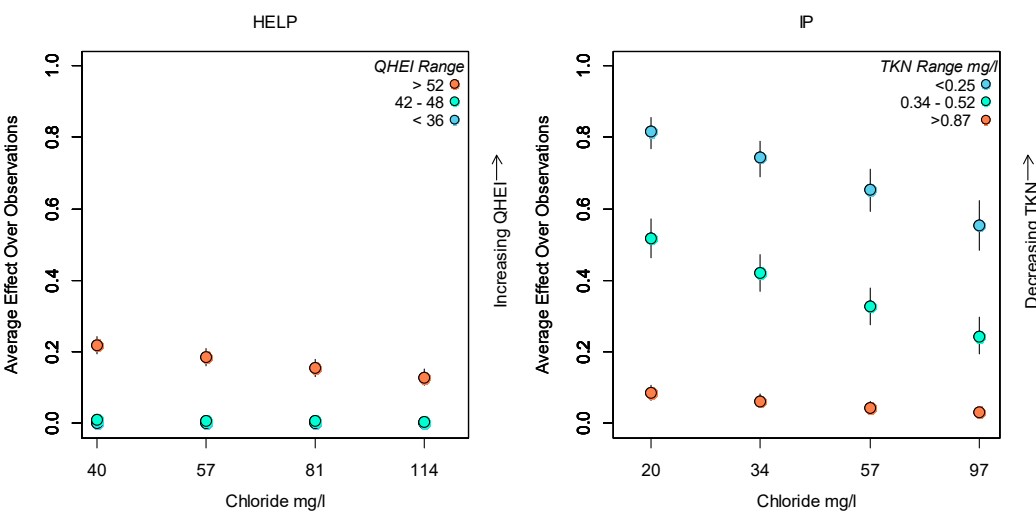

**Figure A1.** Probabilities of observing macroinvertebrate communities meeting the ecoregion benchmarks for the Huron-Erie Lake Plain (HELP, left panel) and the Interior Plateau (IP, right panel) given four levels of chloride and observations of covariates within three data slices defined by quantile levels of the QHEI for the HELP and TKN for the IP (lower, middle and upper 25th quantiles). The respective chloride levels were obtained by sequencing the ecoregion range from the 25th to 90th percentiles (in log space). The results for the HELP illustrate the dramatic influence of poor habitat quality. The results for the IP suggest that ecoregion is particularly susceptible to negative effects from organic and nutrient enrichment.

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
