# Peer review of "Assessing the Impacts of Chloride and Sulfate Ions on Macroinvertebrate Communities in Ohio Streams"

_water, doi:10.3390/w13131815_

Round 1

Reviewer 1 Report

This study has potential to be useful for understanding the impacts of chloride and sulfate ions on macroinvertebrates in streams of the study region and beyond. However, the availability of data used for the study must be clarified and perhaps even added to the manuscript before it will be acceptable for publication. I said the quality of presentation and scientific soundness of this manuscript (in its current form) are "low" due only to a few issues. The vast majority of the work appears to be well-described and sound. However, the data availability issue is a major one.

Most importantly, I could not find a way to access the data set that was analyzed for this study. Usually, these are included at least as supplementary material, or a link or reference is given to a permanent data repository. If access to the data is not given, it must be added in some way. If I simply overlooked it, then it should be made more prominent in the manuscript.

In the taxa sensitivities section, specific taxonomic groups are discussed (e.g., mayflies), but I find no table or listing to see what is being discussed or to verify these statements. This ties back to the previous statement about access to data.

Genus and species names (e.g., line 472) should be italicized.

Define the ecoregion acronyms in the text; Table One and some text spells them out, but I never see all of them directly defined early in the text. Section 4.1 does seem to define some, however. This just needs an all inclusive statement early in the manuscript.

A figure showing a map of the ecoregions of Ohio (as used in this manuscript) would be very useful.

Reviewer 2 Report

Manuscript ID: water-1245899

Title: Assessing the Impacts of Chloride and Sulfate Ions on Macroinvertebrate Assemblages in Freshwater Streams

The manuscript deals with a large amount of data collected throughout the several years on numerous watercourses in different eco-regions. According to that the data set enables interesting and reliable results for readers of the journal as well as for the environmental management.

This large data set could provide clearer results and maybe offer more conclusions to be drawn at the end, if author has stratified the data set and reduced it to a more consistent set and performed additional rounds of analyses.  

Firstly, I suggest to delete all the data form the sites which lack the QHEI (Habitat Quality index) scores. Secondly, I suggest to omit the set of data from the HELP (Huron-Erie Lake Plain) region, for the following reasons:

Figure 1. displays significantly lower quality of riverine habitats (QHEI) in HELP region as well as significantly higher concentration of Total Nitrogen in the mentioned region. It is generally known and quite possible also in this research, that pollution with wastewater, represented as higher TKN values or general degradation of the habitat, represented as QHEI, might have overridden the effects of chloride and sulfate ions which are the focus of this research.

Moreover, the author finds out by himself that poor habitat quality plays greater role than other factors, which is for instance mentioned in Lines 324 and 334.

I suggest slightly different title: Assessing the Impacts of Chloride and Sulfate Ions on Macroinvertebrate Community in Watercourses in Ohio

Please insert a map of the research area and mark the surveyed watercourses.

Minor comments:

Ln.9: I suggest: U.S. standard (instead of national)

Ln.32 and 34: Please use the widely used units for Specific Conductance (µS/cm) elsewhere in the text.

Ln.52: parent and surficial geology – I suggest: geology of the catchment area

Ln.56: least benign – “most threatening”  would sound better

Ln.101-109: Please add a hypothesis to conclude the introduction.

Ln. 128: Macroinvertebrate community

Ln. 145: Caption to Figure 1 should contain explanations of abbreviations.

Table 1. Since this Table is not necessary for understanding I suggest to move it to the Supplementary materials.

Lns. 324 and 334: the author finds out by himself that poor habitat quality plays greater role than other factors.

Ln. 472 write Microtendipes rydalensis in italics

Ln. 545: biological assemblages – I suggest to write: macroinvertebrate community (this is also the focus of this research)

Round 2

Reviewer 2 Report

The author has considered several comments, but has not yet responded to main concerns connected with the consistent data sets. I included these concerns from previous round: 

This large data set could provide clearer results and maybe offer more conclusions to be drawn at the end, if author has stratified the data set and reduced it to a more consistent set and performed additional rounds of analyses. 

I suggest to omit the set of data from the HELP (Huron-Erie Lake Plain) region, for the following reasons:

Figure 1. displays significantly lower quality of riverine habitats (QHEI) in HELP region as well as significantly higher concentration of Total Nitrogen in the mentioned region. It is generally known and quite possible also in this research, that pollution with wastewater, represented as higher TKN values or general degradation of the habitat, represented as QHEI, might have overridden the effects of chloride and sulfate ions which are the focus of this research.

Moreover, the author finds out by himself that poor habitat quality plays greater role than other factors, which is for instance mentioned in Lines 324 and 334.
